

# Effect of adjuvant radioactive iodine therapy on survival in rare oxyphilic subtype of thyroid cancer (Hürthle cell carcinoma)

Qiong Yang[1,*], Zhongsheng Zhao[2,*], Guansheng Zhong[1], Aixiang Jin[1] and Kun Yu[3]

[1] Department of Breast and Thyroid Surgery, Zhejiang Provincial People's Hospital, People's Hospital of Hangzhou Medical College, Hangzhou, Zhejiang, P.R.China
[2] Department of Pathology, Zhejiang Provincial People's Hospital, People's Hospital of Hangzhou Medical College, Hangzhou, Zhejiang, P.R.China
[3] Department of Head, Neck & Thyroid Surgery, Zhejiang Provincial People's Hospital, People's Hospital of Hangzhou Medical College, Hangzhou, Zhejiang, P.R.China
[*] These authors contributed equally to this work.

Corresponding author
Kun Yu, yukun@hmc.edu.cn,
342863441@qq.com

## ABSTRACT

**Purpose**. Radioactive iodine (RAI) is widely used for adjuvant therapy after thyroidectomy, while its value for thyroid cancer has been controversial recently. The primary objectives of this study were to clarify the influence of Radioactive iodine (RAI) on the survival in rare oxyphilic subtype of thyroid cancer (Hürthle cell carcinoma, HCC).

**Methods**. Patients diagnosed with oxyphilic thyroid carcinoma from 2004 to 2015 were extracted from the Surveillance, Epidemiology, and End Results Program database. The Kaplan-Meier method was used to compare overall survival (OS) and cancer-specific survival (CSS) among patients who had adjuvant RAI use or not. Univariate and multivariate Cox proportional hazard models were performed for survival analysis, and subsequently visualized by nomogram.

**Results**. In all, 2,799 patients were identified, of which 1529 patients had adjuvant RAI use while 1,270 patients had not. Based on multivariate Cox analysis, the RAI therapy confers an improved OS for HCC patients (HR = 0.57, 95% CI [0.44–0.72], $P < 0.001$), whereas it has no significant benefit in the survival analysis regarding CSS (HR = 0.79, 95% CI [[0.47–1.34], $P = 0.382$). In a subgroup analysis, the same survival benefit of RAI treatment on OS, but not CSS was observed among patients stratified by AJCC stage and tumor extension. Nevertheless, patients with regional lymph node metastasis benefited from RAI therapy both in OS and CSS ($P < 0.001$, respectively). Furthermore, nomograms used for predicting long term survival of HCC patients exhibited a better prediction power for OS compared with traditional tumor, nodal and metastatic (TNM) stage made by American Joint Committee on Cancer (AJCC) (C-index = 0.833 of the nomogram model vs. 0.696 of the AJCC system).

**Conclusions**. This study suggests that RAI therapy is significantly associated with improved OS in patients with Hürthle cell carcinoma. However, there was no association between treatment with radioiodine and CSS, possibly due to small number of deaths that were related to HCC.

## INTRODUCTION

Oxyphilic thyroid carcinoma, also known as Hürthle cell carcinoma (HCC), is a rare and more aggressive thyroid cancer developed from Hürthle cells. Generally speaking, Hürthle cells, or oxyphilic cells, are large polygonal cells with distinct borders in appearance originated from thyroid follicular epithelium (*Ahmed et al., 2008*). This rare type of thyroid cancer is comparatively less studied and accounts for 3–4% of all thyroid cancers (*Hundahl et al., 2015*). HCC, unlike its benign counterpart called Hürthle cell adenoma, is identified as thyroid malignancy characterized with presence of capsular or vascular invasion, which cannot be determined by fine needle aspiration (FNA) (*Mills et al., 2009*). Moreover, the World Health Organization categorizes HCC as a variant of follicular carcinoma, hinting that HCC exhibits some characteristics similar to those of follicular carcinoma (*Phitayakorn & McHenry, 2006*). Specifically, according to the extent of vascular invasion, HCC can also be subdivided into minimally invasive type and extensive invasive type (*Ganly et al., 2013*). It has been well accepted that HCC have a high propensity for local invasion, lymphatic metastasis to the neck, or distant metastasis. As a consequence, patients suffered from HCC have a poor prognosis in comparison with other differentiated thyroid carcinoma due to its relatively higher recurrence rate and disease-specific mortality rate (*Ghossein et al., 2006*; *Kushchayeva et al., 2004*; *Shaha, Shah & Loree, 1996*).

Radioactive iodine (RAI) therapy is known widely to be effective in the treatment of thyroid cancer after thyroidectomy. However, as thyroid cancer treatment gradually tends to be conservative, the accurate survival benefit of postoperative RAI therapy remains disputable. Recently, there are increasing number of studies focusing on assessing the impact of postoperative RAI therapy on survival of thyroid cancer patient with different risk of recurrence identified by the American Thyroid Association (ATA) risk stratification system, and the results have varied (*Ruel et al., 2015*; *Schvartz et al., 2012*). However, to date, there is no reliable large population based research evaluating whether patients with HCC can benefit from postoperative RAI therapy. Hence, since HCC patients are considered to be refractory to RAI therapy because of its decreased avidity for $I^{131}$ (*Besic et al., 2003*; *Lopez-Penabad et al., 2003*), accurately evaluating the role of RAI therapy on the prognosis of HCC has become increasingly important.

In current study, a large population of HCC patients were identified in the Surveillance, Epidemiology, and End Results (SEER) database. Overall survival (OS) and cancer-specific survival (CSS) were comprehensively compared between patients who had been treated with RAI therapy and those who had not. The purpose of this study was to evaluate whether implementing RAI therapy could exert a clinical benefit in improving survival in patients with oxyphilic thyroid carcinoma.

## MATERIALS & METHODS

### Patient selection and variables extracted

The SEER program is the largest publicly available cancer dataset containing information on cancer incidence, patient characteristics, and survival from various locations throughout the United States, which represents approximately 28% of the US population (*Wu et al., 2017*). In our study, we signed a SEER research data agreement form to access the SEER data. Then, all the information of patients diagnosed with oxyphilic thyroid carcinoma (ICD-O-3 8290/3) from 2004 to 2015 were retrieved from 18 registries Custom Data (with additional treatment fields) using the SEER*Stat program (v 8.3.5). The inclusion criteria was as follows: (a) patients whose complete dates are available (i.e., date last contact ≥ date of diagnosis); (b) radiation recode was "None" or "Radioisotopes"; in addition, patients with unknown surgical type were excluded. Finally, a total of 2799 eligible patients were evaluated in this study. Since the SEER database is public and available worldwide, the informed consent is not required.

Variables including age, race, gender, marital status, surgery type, lymph nodes surgery type, chemotherapy recode, extension, AJCC TNM classification and tumor size, SEER cause-specific death classification, vital status and survival months were extracted and assessed. OS was defined as the date from diagnosis to death for any cause. CSS was defined as that the death were attributed directly to oxyphilic thyroid carcinoma.

### Statistical analysis

In this study, age was divided into < 45, 45–64 and ≥ 65 years respectively. Marital status was coded into two categories: married and not married (single, widowed, separated and divorced). Since the detection rate of papillary thyroid microcarcinoma (PTMC) is increasing continually, partly due to rapid development of imaging diagnostic techniques, we redivided HCC into three subgroup (≤10 mm, 11–39 mm and ≥40 mm) according to tumor size. A chi-square test was utilized to assess the baseline characteristics between RAI and no RAI group. Univariate and multivariate logistic regression analysis were used to identify factors associated with treatment of radioiodine. Kaplan–Meier plot and log-rank test were used to compare differences of OS and CSS between RAI and no-RAI group. Univariate and multivariate Cox proportional hazard model were performed for survival analysis in OS and CSS, and subsequently visualized by nomogram. All the statistical analyses were performed using SPSS 24.0 (Chicago, IL, USA). All the survival curve and nomogram were plotted using R software version 3.4.4. A two-tailed $P < 0.05$ was considered statistically significant.

## RESULTS

### Demographic characteristics of patients

According to the inclusion criteria, a total of 2,799 patients diagnosed with Hürthle cell carcinoma were identified from the SEER database, of which 1,529 patients received adjuvant RAI therapy while 1,270 patients did not receive RAI therapy. The baseline characteristics in each group are summarized in Table 1. As shown in Table 1, patients in

**Table 1  Baseline characteristics of HCC patients in this study.**

| Characteristics | Total, *n* (%) N = 2,799 | No RAI, *n* (%) N = 1,270 | RAI, *n* (%) N = 1,529 | * *P* value |
|---|---|---|---|---|
| Age | | | | |
| <45 | 587 (21) | 240 (18.9) | 347 (22.7) | <0.001 |
| 45–64 | 1,207 (43.1) 515 (40.6) | 692 (45.3) | | |
| ≥ 65 | 1,005 (35.9) | 515 (40.6) | 490 (32 ) | |
| Race | | | | |
| Black | 240 (8.6) | 107 (8.4) | 133 (8.7) | 0.678 |
| White | 2341 (83.6) | 1,058 (83.3) | 1,283 (83.9) | |
| Other | 218 (7.8) | 105 (8.3) | 113 (7.4) | |
| Sex | | | | |
| Female | 1,938 (69.2) | 902 (71) | 1,036 (67.8) | 0.062 |
| Male | 861 (30.8) | 368 (29) | 493 (32.2) | |
| Marital status | | | | |
| Married | 1,658 (59.2) | 714 (56.2) | 944 (61.7) | 0.001 |
| Not married | 990 (35.4) | 468 (36.9) | 522 (34.1) | |
| Unknown | 151 (5.4) | 88 (6.9) | 63 (4.1) | |
| Surgery | | | | |
| Total thyroidectomy | 2,040 (72.9) | 731 (57.6) | 1,309 (85.6) | <0.001 |
| Partial thyroidectomy | 668 (23.9) | 450 (35.4) | 218 (14.3) | |
| No surgery | 91 (3.3) | 89 (7.0) | 2 (0.1) | |
| Lymph nodes surgery | | | | |
| No | 1,903 (68.0) | 938 (73.9) | 965 (63.1) | <0.001 |
| Regional lymph nodes dissection | 822 (29.4) | 291 (22.9) | 531 (34.7) | |
| Unknown | 74 (2.6) | 41 (3.2) | 33 (2.2) | |
| Chemotherapy | | | | |
| No | 2,784 (99.5) | 1,259 (99.1) | 1,525 (99.7) | 0.029 |
| Chemotherapy | 15 (0.5) | 11 (0.9) | 4 (0.3) | |
| Seer_stage | | | | |
| Localized | 2,347 (83.9) | 1,049 (82.6) | 1,298 (84.9) | <0.001 |
| Regional | 305 (10.9) | 130 (10.2) | 175 (11.4) | |
| Distant | 98 (3.5) | 50 (3.9) | 48 (3.1) | |
| Unknown | 49 (1.8) | 41 (3.2) | 8 (0.5) | |
| AJCC_stage | | | | |
| I | 1,077 (38.5) | 486 (38.3) | 591 (38.7) | <0.001 |
| II | 686 (24.5) | 304 (23.9) | 382 (25.0) | |
| III | 687 (24.5) | 276 (21.7) | 411 (26.9) | |
| IV | 163 (5.8) | 79 (6.2) | 84 (5.5) | |
| Unknown | 186 (6.6) | 125 (9.8) | 61 (4.0) | |
| T | | | | |
| T1 | 676 (24.2) | 338 (26.6) | 338 (22.1) | <0.001 |
| T2 | 962 (34.4) | 415 (32.7) | 547 (35.8) | |

*(continued on next page)*

**Table 1** (*continued*)

| Characteristics | Total, *n* (%) N = 2,799 | No RAI, *n* (%) N = 1,270 | RAI, *n* (%) N = 1,529 | * *P* value |
|---|---|---|---|---|
| T3 | 908 (32.4) | 359 (28.3) | 549 (35.9) | |
| T4 | 95 (3.4) | 45 (3.5) | 50 (3.3) | |
| TX | 158 (5.6) | 113 (8.9) | 45 (2.9) | |
| N | | | | |
| N0 | 2,540 (90.7) | 1,125 (88.6) | 1,415 (92.5) | <0.001 |
| N1 | 141 (5.0) | 60 (4.7) | 81 (5.3) | |
| NX | 118 (4.2) | 85 (6.7) | 33 (2.2) | |
| M | | | | |
| M0 | 2,657 (94.9) | 1,182 (93.1) | 1,475 (96.5) | <0.001 |
| M1 | 67 (2.4) | 30 (2.4) | 37 (2.4) | |
| MX | 75 (2.7) | 58 (4.6) | 17 (1.1) | |
| Tumor_size (mm) | | | | |
| ≤10 | 184 (6.6) | 130 (10.2) | 54 (3.5) | <0.001 |
| 11–39 | 1,580 (56.4) | 676 (53.2) | 904 (59.1) | |
| ≥40 | 859 (30.7) | 341 (26.9) | 518 (33.9) | |
| Unknown | 176 (6.3) | 123 (9.7) | 53 (3.5) | |
| Extension | | | | |
| Intrathyroidal | 1,499 (53.6) | 699 (55.0) | 800 (52.3) | <0.001 |
| Extrathyroidal | 1,131 (40.4) | 463 (36.5) | 668 (43.7) | |
| Unknown | 169 (6.0) | 108 (8.5) | 61 (4.0) | |

**Notes.**

\* *P* values were calculated by chi-square test.

the RAI group had higher proportion of 45-64-year-old age (45.3% vs 40.6%) and marriage (61.7% vs 56.2%) compared with no-RAI group. In addition, patients in RAI group tended to have tumor being larger, later AJCC stage and more extent of extrathyroidal invasion than those in the no-RAI group. Moreover, the RAI group had a higher percentage of total thyroidectomy (85.6% vs 57.6%) and regional lymph nodes dissection (34.7% vs 22.9%) than no-RAI group. The surprising thing is that there are 7% percentage of patients in no-RAI group who did not receive surgery treatment. Taking into consideration that HCC is only diagnosed in histologic analysis (*Mills et al., 2009*), its diagnosis is most likely to come from the autopsy according to surgery codes of thyroid gland supplied by SEER program.

## Factors associated with use of radioactive iodine therapy

For the purpose of better understanding the clinicopathological factors associated with implementing of RAI therapy, univariate and multivariate logistic regression analysis were performed (Table 2). In univariate analysis, we demonstrated that age ≥ 65, not married, and patients received partial thyroidectomy or no surgery were less likely of receiving RAI therapy. Patients who underwent regional lymph nodes dissection and had tumors with aggressive characteristics, such as larger tumor size and more extrathyroidal involvement, were significantly associated with increased propensity of RAI use. Moreover, the multivariate logistic analysis further suggested that patients who were younger age,

**Table 2  Factors associated with RAI use.**

| | Univariate logistic model | | Multivariate logistic model | |
| --- | --- | --- | --- | --- |
| Variables | OR (95% CI) | *P* value | OR (95% CI) | *P* value |
| Age | | | | |
| <45 | Reference | | Reference | |
| 45–64 | 0.93 (0.76–1.14) | 0.47 | 0.95 (0.76–1.18) | 0.62 |
| ≥ 65 | 0.66 (0.54–0.81) | <0.001 | 0.71 (0.56–0.88) | 0.002 |
| Sex | | | | |
| Female | Reference | | Reference | |
| Male | 1.17 (0.99–1.37) | 0.062 | 1.13 (0.94–1.36) | 0.197 |
| Marital status | | | | |
| Married | Reference | | Reference | |
| Not married | 0.84 (0.72–0.99) | 0.035 | 0.92 (0.78–1.10) | 0.38 |
| Surgery | | | | |
| Total thyroidectomy | Reference | | Reference | |
| Partial thyroidectomy | 0.27 (0.23–0.33) | <0.001 | 0.28 (0.23–0.33) | <0.001 |
| No surgery | 0.01 (0.003–0.05) | <0.001 | 0.02 (0.005–0.09) | <0.001 |
| Lymph nodes surgery | | | | |
| No | Reference | | Reference | |
| Regional lymph nodes dissection | 1.77 (1.50–2.10) | <0.001 | 1.50 (1.25–1.80) | <0.001 |
| Tumor size (mm) | | | | |
| ≤ 10 | Reference | | Reference | |
| 11–39 | 3.22 (2.31–4.49) | <0.001 | 3.50 (2.46–4.97) | <0.001 |
| ≥ 40 | 3.66 (2.59–5.17) | <0.001 | 4.16 (2.86–6.04) | <0.001 |
| Lymph nodes metastases | | | | |
| No | Reference | | | |
| Yes | 1.073 (0.76–1.51) | 0.686 | | |
| Distant metastases | | | | |
| No | Reference | | | |
| Yes | 0.99 (0.61–1.61) | 0.61 | | |
| Extension | | | | |
| Intrathyroidal | Reference | | Reference | |
| Extrathyroidal | 1.26 (1.08–1.47) | 0.004 | 1.14 (0.96–1.35) | 0.086 |

**Notes.**
OR, odds ratio.

larger tumor size, received total thyroidectomy or regional lymph nodes dissection were more liable to be treated with RAI therapy.

## Survival analysis of all population

The median follow-up duration in this study was 5.2 years (mean, 5.4 years; range, 0 to 12 year). Of all the 2,799 patients finally evaluated, a total of 337 patients were dead at the time of last follow-up. Importantly, 85 patients were dead directly from oxyphilic thyroid carcinoma. Subsequently, the Kaplan–Meier plots was utilized to compare the overall survival (OS) and cancer specific survival (CSS) between patients who had, or had not received RAI therapy. As shown in Fig. 1, patients received RAI therapy had better OS

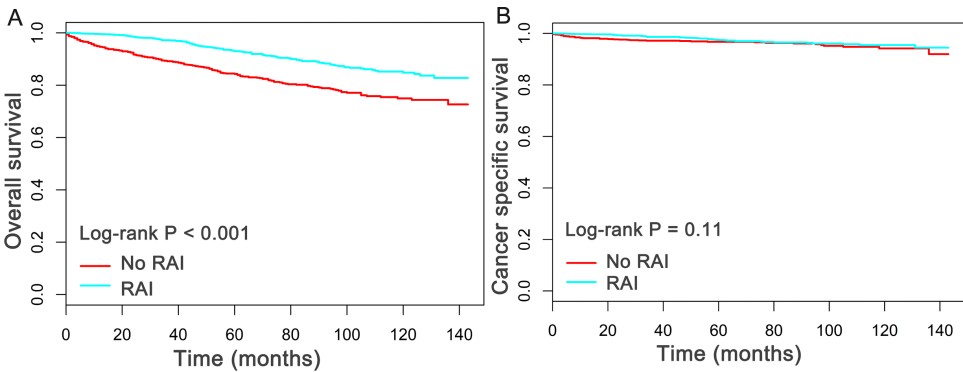

**Figure 1** Overall survival (OS) and Cancer-specific survival (CSS) curves plotted by Kaplan-Meier method for patients received RAI therapy or not.

($P < 0.05$) than patients who had not received RAI therapy, whereas there is no statistical significance for CSS between the two group ($P = 0.11$).

Cox proportional models were applied to assess the multiple factors related to survival. As shown in Table 3, the univariate analysis displayed that older age, male, not married, advanced AJCC TNM stage, larger tumor size ($\geq 40$ mm) and extrathyroidal invasion were poor prognostic factors for OS of patients (HR > 1, $P < 0.05$), whereas treatments with RAI, total thyroidectomy and regional lymph nodes dissection were proved to be protective factors for better OS (HR < 1, $P < 0.05$). To consider the possible bias between two group, all the aforementioned variables were then enlisted into the multivariate analysis (Table 3). Notably, RAI use still exerted as protective factor for improved OS (HR = 0.57, 95% CI [0.44–0.72], $P < 0.05$), compared with no-RAI group. Importantly, both univariate and multivariate Cox regression analysis of CSS indicated that patients received RAI therapy has no significant survival advantage over the patients without use of RAI (HR = 0.79, 95% CI [0.44–1.34], $P > 0.05$). The detailed Cox analysis for CSS were shown in Table 4. Taken together, these results suggested that patients with oxyphilic thyroid carcinoma could benefit from RAI therapy in OS, but not in CSS.

## Nomograms for OS and CSS

In order to predict the long term survival rate for patients with oxyphilic thyroid carcinoma, prognostic nomograms were depicted by eight prognostic factors, including age, sex, marital status, surgery type, RAI therapy, AJCC stage, lymph nodes metastatic status and distant metastatic status. Each factors mentioned above were proved to be significantly associated with OS in multivariate Cox proportional hazard regression model. As shown in Fig. 2A, each factors in the nomogram model could be ascribed a weighted point according to the point scale. The sum of the points could be used to predict the long term survival (3 year, 5 year and 10 year overall survival rate) according to the total points axis, wherein a higher score was deemed to have worse prognosis. Similarly, the nomogram employing aforementioned covariates was used to predict the cancer specific survival (Fig. 2B). Subsequently, the C-index for the multifactorial nomogram and the

**Table 3  Cox proportional hazards regression model analysis of overall survival (OS).**

| Characteristics | Univariate cox | | Multivariate cox | |
|---|---|---|---|---|
| | HR (95% CI) | P value | HR (95% CI) | P value |
| Age | | | | |
| <45 | Reference | | Reference | |
| 45–64 | 3.88 (1.94–7.76) | <0.001 | 3.08 (1.43–6.63) | 0.004 |
| ≥ 65 | 21.1 (10.8–40.9) | <0.001 | 13.3 (6.32–27.8) | <0.001 |
| Sex | | | | |
| Female | Reference | | Reference | |
| Male | 1.68 (1.35–2.09) | <0.001 | 1.43 (1.14–1.81) | 0.002 |
| Marital status | | | | |
| Married | Reference | | Reference | |
| Not married | 1.73 (1.39–2.15) | <0.001 | 1.73 (1.37–2.18) | <0.001 |
| Surgery | | | | |
| Total thyroidectomy | Reference | | Reference | |
| Partial thyroidectomy | 1.40 (1.10–1.79) | 0.007 | 1.14 (0.88–1.48) | 0.312 |
| No surgery | 7.46 (5.29–10.5) | <0.001 | 4.03 (2.43–6.70) | <0.001 |
| Lymph nodes surgery | | | | |
| No | Reference | | Reference | |
| Regional lymph nodes dissection | 0.71 (0.55–0.92) | 0.009 | 0.82 (0.60–1.11) | 0.201 |
| Radiation | | | | |
| No RAI | Reference | | Reference | |
| RAI | 0.47 (0.37–0.58) | <0.001 | 0.57 (0.44–0.72) | <0.001 |
| AJCC stage | | | | |
| I | Reference | | Reference | |
| II | 2.62 (1.84–3.75) | <0.001 | 1.32 (0.88–1.98) | 0.178 |
| III | 3.69 (2.63–5.18) | <0.001 | 1.07 (0.64–1.80) | 0.794 |
| IV | 14.9 (10.4–21.3) | <0.001 | 2.77 (1.52–5.05) | 0.001 |
| Tumor size (mm) | | | | |
| ≤ 10 | Reference | | Reference | |
| 11–39 | 1.26 (0.72–2.23) | 0.420 | 1.11 (0.60–2.07) | 0.738 |
| ≥ 40 | 2.30 (1.30–4.06) | 0.004 | 1.58 (0.79–3.14) | 0.193 |
| Lymph nodes metastases | | | | |
| No | Reference | | Reference | |
| Yes | 3.46 (2.47–4.84) | <0.001 | 1.65 (1.02–2.69) | 0.043 |
| Distant metastases | | | | |
| No | Reference | | Reference | |
| Yes | 10.39 (7.39–14.6) | <0.001 | 2.31 (1.38–3.86) | 0.002 |
| Extension | | | | |
| Intrathyroidal | Reference | | Reference | |
| Extrathyroidal | 1.44 (1.15–1.80) | 0.002 | 1.10 (0.86–1.40) | 0.442 |

**Table 4  Cox proportional hazards regression model analysis of Cancer-specific survival (CSS).**

| Characteristics | Univariate cox | | Multivariate cox | |
|---|---|---|---|---|
| | OR (95% CI) | *P* value | OR (95% CI) | *P* value |
| Age | | | | |
| <45 | Reference | | Reference | |
| 45–64 | 2.64 (0.91–7.69) | 0.075 | 0.36 (0.06–1.99) | 0.239 |
| ≥ 65 | 10.7 (3.88–29.4) | <0.001 | 0.91 (0.17–4.99) | 0.915 |
| Sex | | | | |
| Female | Reference | | Reference | |
| Male | 1.96 (1.27–3.00) | 0.002 | 1.35 (0.84–2.17) | 0.217 |
| Marital status | | | | |
| Married | Reference | | Reference | |
| Not married | 1.48 (0.96–2.27) | 0.074 | 1.50 (0.93–2.41) | 0.098 |
| Surgery | | | | |
| Total thyroidectomy | Reference | | Reference | |
| Partial thyroidectomy | 0.97 (0.56–1.67) | 0.911 | 1.14 (0.63–2.06) | 0.671 |
| No surgery | 9.48 (5.25–17.1) | <0.001 | 2.88 (1.12–7.41) | 0.028 |
| Lymph nodes surgery | | | | |
| No | Reference | | Reference | |
| Regional lymph nodes dissection | 1.44 (0.91–2.28) | 0.119 | 0.99 (0.54–1.84) | 0.98 |
| Radiation | | | | |
| No RAI | Reference | | Reference | |
| RAI | 0.71 (0.46–1.09) | 0.113 | 0.79 (0.47–1.34) | 0.382 |
| AJCC stage | | | | |
| I | Reference | | Reference | |
| II | 2.44 (0.69–8.65) | 0.167 | 4.48 (0.73–27.6) | 0.106 |
| III | 9.96 (3.44–28.8) | <0.001 | 7.10 (1.21–41.6) | 0.030 |
| IV | 109.3 (39.3–303.5) | <0.001 | 37.4 (6.11–228.7) | <0.001 |
| Tumor size (mm) | | | | |
| ≤ 10 | Reference | | Reference | |
| 11–39 | 1.21 (0.29–5.15) | 0.796 | 1.00 (0.19–5.27) | 0.999 |
| ≥ 40 | 5.46 (1.33–22.5) | 0.019 | 2.20 (0.44–11.0) | 0.339 |
| Lymph nodes metastases | | | | |
| No | Reference | | Reference | |
| Yes | 11.3 (6.91–18.3) | <0.001 | 1.87 (0.90–3.88) | 0.092 |
| Distant metastases | | | | |
| No | Reference | | Reference | |
| Yes | 36.6 (22.8–58.6) | <0.001 | 3.25 (1.66–6.34) | 0.001 |
| Extension | | | | |
| Intrathyroidal | Reference | | Reference | |
| Extrathyroidal | 3.95 (2.35–6.66) | <0.001 | 1.67 (0.95–2.95) | 0.074 |

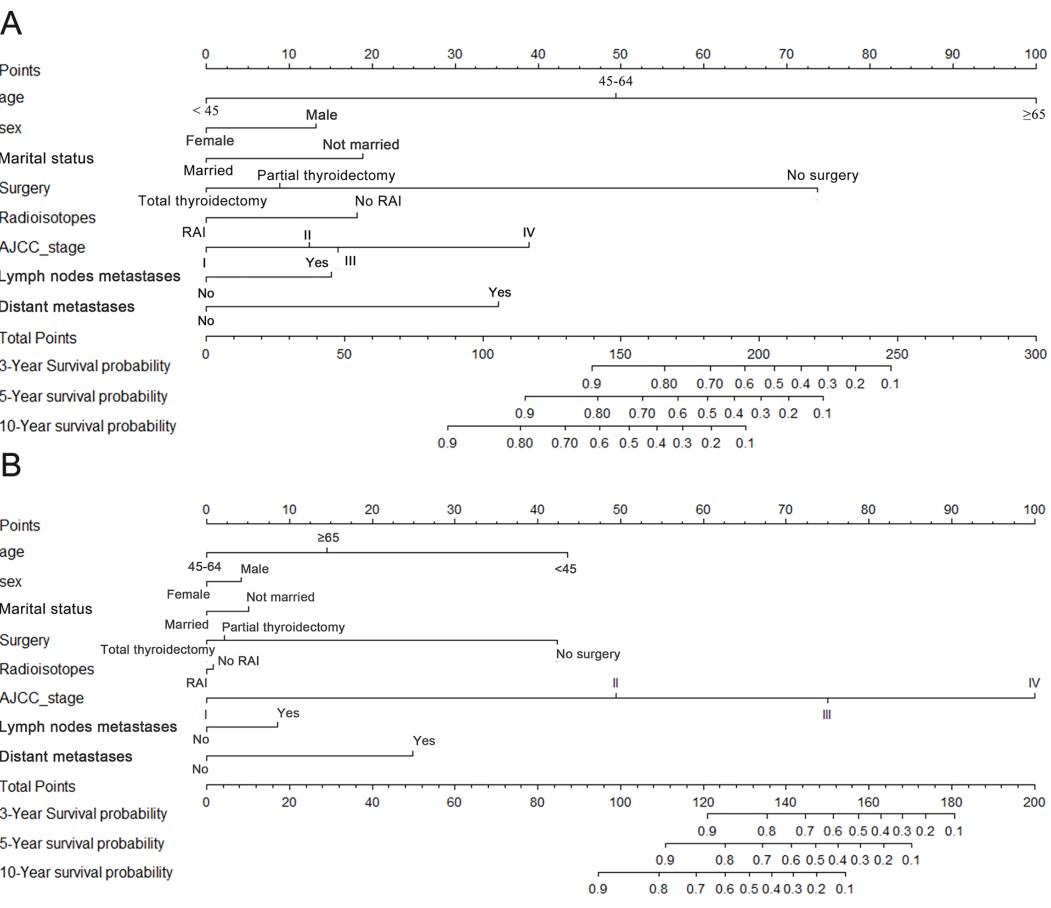

**Figure 2** **Nomograms for (A) OS and (B) CSS.** Nomogram predicting 3- to 10-year survival probability using eight available clinical characteristics.

traditional AJCC TNM staging system were calculated and compared to evaluate the predictive accuracies. For OS and CSS, the c-index were 0.833 and 0.882 respectively, which were higher than that of the traditional AJCC staging system (0.696 and 0.867 respectively). These results, taken together, suggested that the nomograms built in this study were better than the traditional AJCC stage system at predicting prognosis.

## Impact of radioactive iodine on survival in subgroup

In order to better understand the influence of RAI therapy on prognosis for the oxyphilic thyroid carcinoma patients, we further studied the patients' OS and CSS in subgroup stratified by AJCC stage, extension and lymph nodes metastatic status. Regarding AJCC stage, the Kaplan–Meier plots showed significant OS ($P < 0.05$) difference with RAI use in stages I-IV, whereas there is no significant difference in CSS ($P > 0.05$) (Fig. 3). In addition, similar results were shown in subgroup of patients who had oxyphilic thyroid carcinoma with or without extrathyroidal invasion (Fig. 4). Furthermore, RAI therapy also acted a significant survival benefit in OS for patients with no regional lymph node metastasis, whereas there was no significant survival difference in CSS curves (Fig. 5).

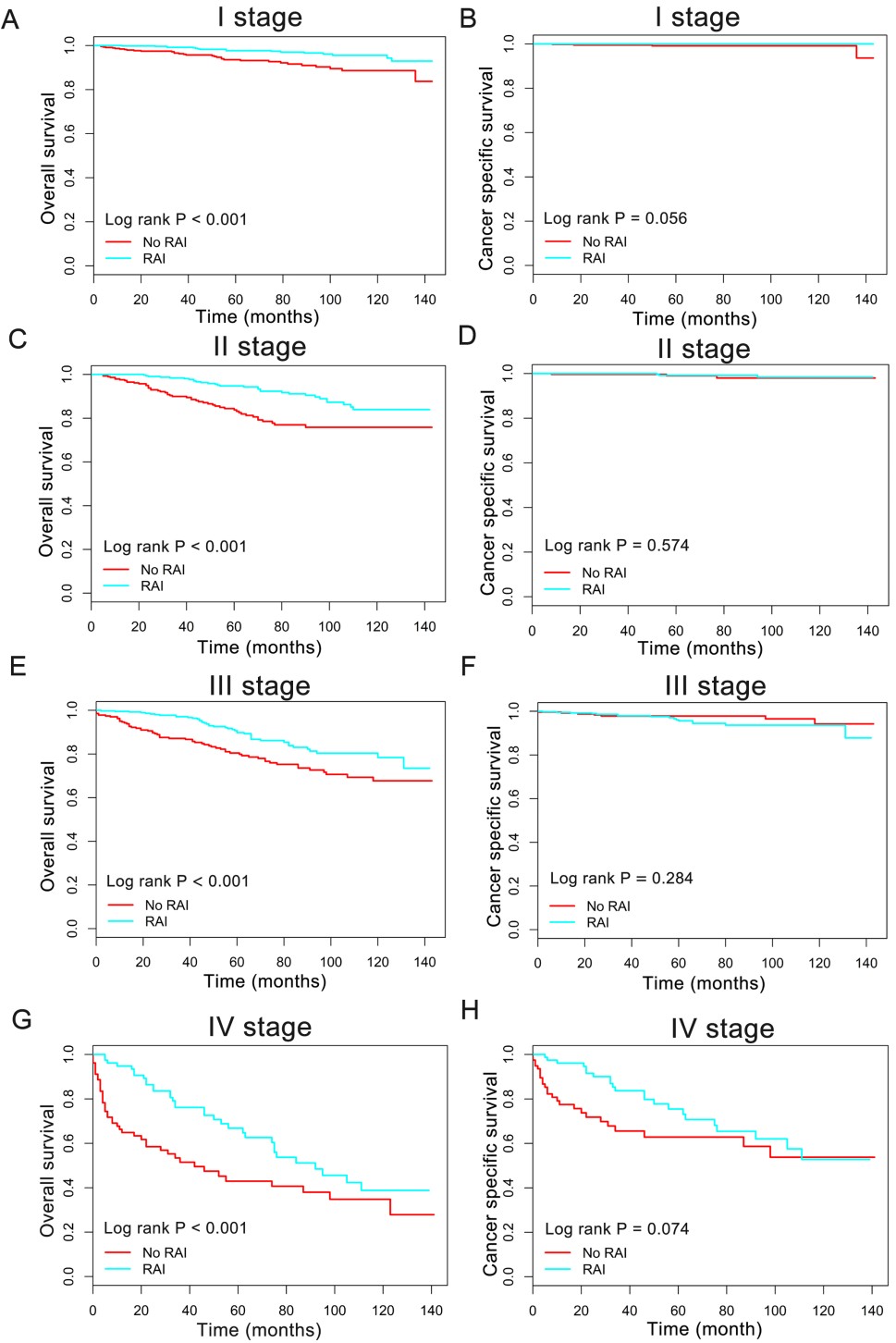

**Figure 3** Overall survival (OS) and cancer-specific survival (CSS) curves plotted by Kaplan-Meier method for I stage (A, B), II stage (C, D), III stage (E, F) and IV stage (G, H) patients.

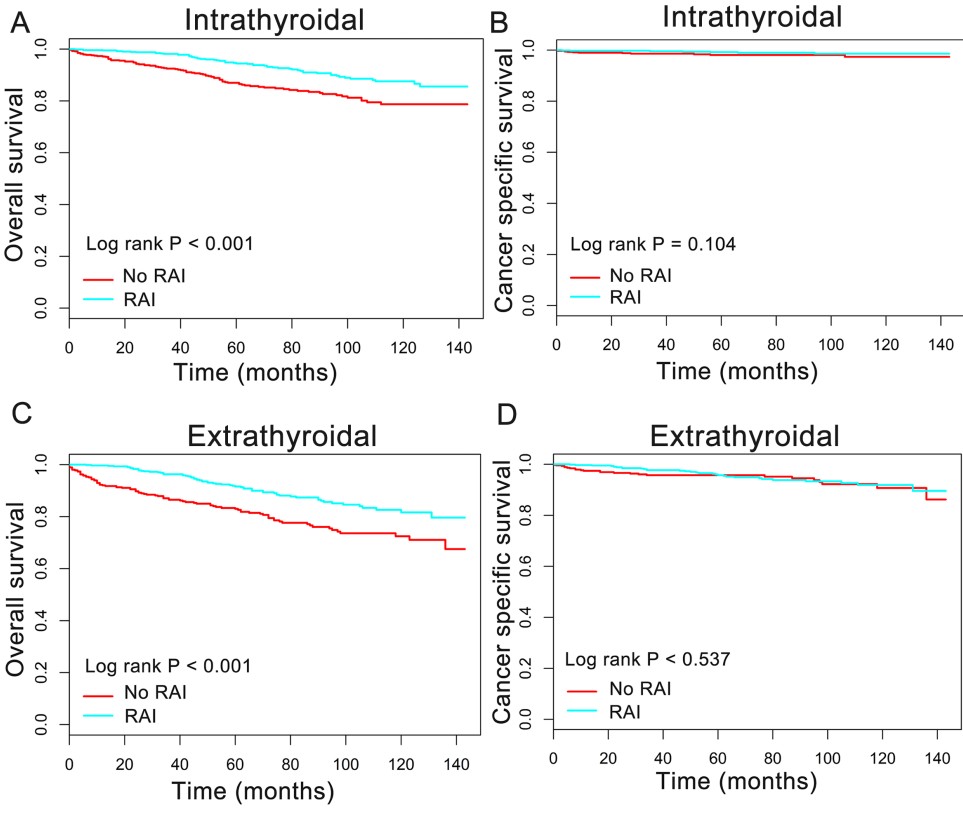

**Figure 4  Overall survival (OS) and cancer-specific survival (CSS) curves plotted by Kaplan-Meier method for patients with or without extrathyroidal invasion.** HCC confined to the thyroid gland (A, B) and HCC with extrathyroidal invasion (C, D).

However, in the subgroups of patients who had regional lymph node metastasis, RAI use has shown a significant survival advantage both in OS and CSS (Fig. 5). Taken together, these data indicated that RAI therapy has significant survival benefits in OS for the whole oxyphilic thyroid carcinoma patients, whereas there seems to be no association between RAI treatment and CSS. Importantly, RAI therapy should be highly recommended for patients who had regional lymph node metastasis because of the survival benefit both in OS and CSS.

## DISCUSSION

HCC, a rare histological type among differentiated thyroid carcinoma, is diagnosed mainly based on postoperative pathological examination, which is characterized with a finely granular eosinophilic cytoplasm with increased number of mitochondria in it (*Montone, Baloch & LiVolsi, 2008*). HCC is classified to be oxyphilic variants of follicular thyroid cancer (FTC), and inconsistent results were reported by previous studies regarding the long term survival of HCCs compared with other differentiated thyroid carcinoma (*Haigh & Urbach, 2005*). For example, some studies have proposed that HCC exhibits more aggressive and poor prognosis than conventional papillary carcinoma and

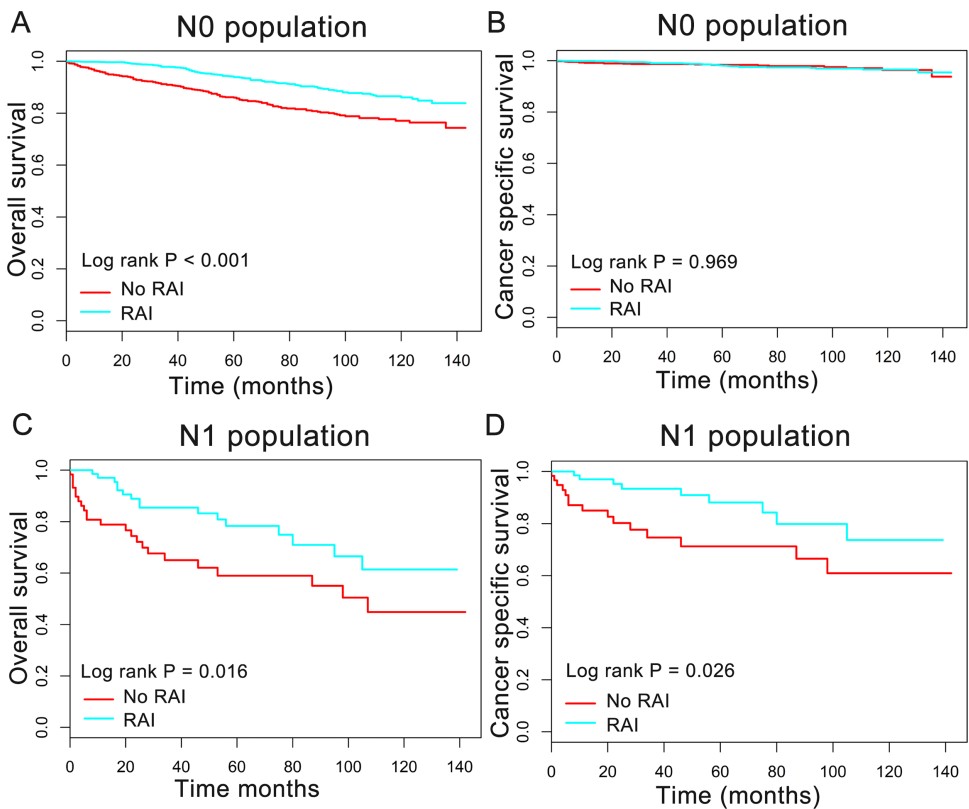

**Figure 5** Overall survival (OS) and cancer-specific survival (CSS) curves plotted by Kaplan-Meier method for N0 (A, B) and N1 (C, D) population among HCC patients.

follicular carcinoma, which should be categorized as distinct tumor (*Azadian et al., 1995*; *Bishop et al., 2012*; *Kushchayeva et al., 2004*; *Kushchayeva et al., 2008*). However, *Haigh & Urbach (2005)* have reported that HCC patients had similar prognosis compared to their non-HCC counterparts after matching important prognostic variables including age and tumor stage, and suggested that patients with HCCs should be treated the same as non-HCC patients with equivalent stage. Generally, a total thyroidectomy is recommended for HCCs due to its aggressive behavior and also due in part to an increased rate of locoregional recurrence (*Bishop et al., 2012*; *Haigh & Urbach, 2005*; *Petric, Gazic & Besic, 2014*).

Recently, RAI therapy is widely used for adjuvant treatment after total thyroidectomy for the purpose of ablating unresectable residual lesions and meanwhile predicting tumor persistence or recurrence by monitoring the thyroglobulin (Tg) level and/or implementing of RAI whole-body scan (*Besic et al., 2003*; *Foote et al., 2003*). However, the recommendations regarding the RAI use for HCC differs across the American Thyroid Association (ATA) and the NCCN guidelines (*Haugen et al., 2016*; *Jillard et al., 2016*). It remains contradictory whether the use of adjuvant RAI therapy might improve survival of patients suffered from HCCs, because it is generally believed that fewer than 10% of HCCs take up radioiodine (*Arganini et al., 1986*; *Kushchayeva et al., 2004*; *Stojadinovic et al., 2002*). To the best of our knowledge, there is no large clinical trials that evaluated the

efficacy or survival benefit of RAI use for HCC except for a few small retrospective studies (*Kushchayeva et al., 2004*; *Lopez-Penabad et al., 2003*), largely because of its relative rarity. Therefore, multi-institutional trials should be pursued to standardize and reinforce the published management guidelines.

In this study, we retrospectively analyzed the difference between two cohort of HCC patients received RAI treatment or not, based on a large population extracted from SEER database. We observed that patients who received RAI therapy tend to have tumors with later stage and receive more aggressive treatment like total thyroidectomy or regional lymph nodes dissection. Intriguingly, a higher proportion of patients who had married were observed in the RAI group. We also assessed the clinicopathological variables significantly associated with use of RAI therapy using the univariate and multivariate logistic regression analysis. Our result revealed that patients with younger age, larger tumor size and received total thyroidectomy or regional lymph nodes dissection were independent factors associated with use of RAI.

In order to evaluate whether adjuvant RAI therapy had an improved survival for HCCs patient, we performed survival analysis using Kaplan–Meier plots and Cox analysis. We demonstrated that the RAI therapy confers an improved OS for HCCs patients after taking into consideration of multiple clinicopathological variables in multivariate Cox's regression analysis. Interestingly, both univariate and multivariate Cox survival analysis showed that patients who have married had better OS than unmarried one. We hypothesized that either financial or psychological support from family could be an important contributing factors associated with good prognosis. Nevertheless, in the survival analysis regarding CSS, the result showed that no survival benefit of RAI therapy was apparent both in univariate and multivariate Cox proportional hazard regression model. However, of the totally 2,799 patients, only 85 cases were dead directly from Hürthle cell carcinoma, hence we can't conclude that HCCs patients didn't achieve survival benefits from implement of RAI therapy due to the insufficient statistical power of incidents. Further validation cohort or large prospective studies containing more patients death attributed directly to HCCs are needed in the future.

In our study, specific nomogram prediction models for OS and CSS were developed separately for patients with Hürthle cell carcinoma. Based on the results from multivariate Cox regression analysis, we utilized eight factors significantly associated with overall survival to predict long term survival in the nomogram model, in which a higher score point suggested worse prognosis. Subsequently, we evaluated the predictive accuracy of the nomogram model and compared to that of the traditional AJCC staging system though calculating the C-index. The results indicated that the nomogram model regarding OS showed a better prognosis power than traditional AJCC TNM staging system (C-index = 0.833 of the nomogram model vs. 0.696 of the AJCC stage system). So, our nomogram system appears to be a better alternative to predict long term survival in patients suffered from Hürthle cell carcinoma.

In order to evaluate the influence of RAI treatment on prognosis of HCC patients, we further stratified all the population for further survival analysis according to AJCC stage, status of regional lymph nodes and extent of extrathyroidal invasion. In consequence,

we came to the same conclusion in subgroup survival analysis for patients both in I–II and III–IV population, and patients with or without extrathyroidal invasion. Notably, the subgroup analysis for patients suffered regional lymph node metastasis indicated that RAI therapy exerted survival benefits both in OS and CSS. As lymph node metastases are reportedly more common in patients with HCC (*Evans & Vassilopoulou-Sellin, 1998*), our result highlighted the vital role of RAI therapy for HCC patients with regional lymph node metastasis

Inevitably, the current study has some limitations. Firstly, the SEER database lacked information regarding cancer recurrence, which limited us to assess the impact of RAI treatment on locoregional recurrence. Secondly, the information, including dosage of RAI, extranodal extension and size of the largest metastatic lymph node were not available from SEER database, which were reported to have potential to affect the prognosis of thyroid cancer (*Randolph et al., 2012*; *Wu et al., 2015*). Thirdly, well-known thyroid cancer oncogenes, such as *BRAF*, were not available or adjusted in our study. Furthermore, the number of events occurred in CSS analysis were too small and thus compromised the statistical power.

## CONCLUSIONS

In summary, this study suggests that RAI therapy was significantly associated with improved OS in patients with Hürthle cell carcinoma. However, there was no association between treatment with radioiodine and CSS, possibly due to small number of deaths that were related to HCC.

## ACKNOWLEDGEMENTS

This study is based on the SEER database.

### Funding
This work was financially supported by grant Y201840485 from the General Scientific Research Project of Zhejiang Education Department. The funders had no role in study design, data collection and analysis, decision to publish, or preparation of the manuscript.

### Grant Disclosures
The following grant information was disclosed by the authors:
General Scientific Research Project of Zhejiang Education Department: Y201840485.

### Competing Interests
The authors declare there are no competing interests.

### Author Contributions
- Qiong Yang and Aixiang Jin analyzed the data, contributed reagents/materials/analysis tools, authored or reviewed drafts of the paper.

- Zhongsheng Zhao analyzed the data, contributed reagents/materials/analysis tools, prepared figures and/or tables, authored or reviewed drafts of the paper.
- Guansheng Zhong analyzed the data, contributed reagents/materials/analysis tools, prepared figures and/or tables, authored or reviewed drafts of the paper, approved the final draft.
- Kun Yu conceived and designed the experiments, contributed reagents/materials/analysis tools.

## Data Availability

The raw data of Hürthle cell carcinoma are available in the Supplemental File.

## Supplemental Information

Supplemental information for this article can be found online at http://dx.doi.org/10.7717/peerj.7458#supplemental-information.

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
