# Peer review of "Effect of adjuvant radioactive iodine therapy on survival in rare oxyphilic subtype of thyroid cancer (Hürthle cell carcinoma)"

_PeerJ, doi:10.7717/peerj.7458_

## Round 0.1 · original submission · Major Revisions

Dear Dr. Guansheng,

As you will see, the reviewers have suggested a series of modifications in order to improve your manuscript.

Please answer to all the points raised by the three reviewers in a point-by-point letter.

[]

·

Basic reporting

- The paper states important findings, but needs lot of revision from grammatical standpoint to be considered for "pass" state
- Sentences throughout the paper need to be paraphrased because of grammatical errors
- In discussion section, would suggest further comprehensive research into what is being done to address this issue from future standpoint. Are there any clinical trials to address this?
- Also, figures for OS and CSS combined stage I-II and III-IV, though all stages have different course, as a result would suggest looking at each stage individually to confer exact OS and CSS difference.

Experimental design

- Original and primary research which is addressed within the Aims and Scope of the journal
- Research question well defined, followed by clear evidence in results supporting it.
- Rigorous investigation performed and results stated clearly with the help of table and figures, with valid statistical analysis.
- Methods described well, with sufficient detail and information to replicate.

Validity of the findings

- All underlying data has been provided along with valid tables and figures
- Conclusion needs to be paraphrased like other sections given grammatical errors

Additional comments

Dear Drs. Yang & Zhao,

It was a pleasure reviewing your article on HCC. You stated some new findings, which need to be further looked at in prospective trials to further validate your results. However, before the paper is considered for publication I believe it needs major revision from grammatical standpoint.

There are lot of grammatical errors, as a result sentences need to be paraphrased. In addition to that, I would advise looking at all the AJCC stages separately - stage I, II, III, IV to evaluate the OS and CSS. Also, in the discussion section, would recommend highlighting what is being done at this point from future perspective, that is stating any clinical trials looking at this question, or what is the best way moving forward with this issue.

Regards,
PeerJ Reviewer

Reviewer 2 ·

Basic reporting

1- Use of English is of poor quality throughout the text and needs to be comprehensively revised. If some of the problems are just simple grammar misuse, other sentences are not understandable. Here are a few examples: «receipt of radiotherapy» (line 98); sentence in lines 110-112 need to be rewritten because it is difficult to understand; «Extrathyroidal evasion» (line 114). «At the end of the last follow-up» (lines 130-131); «Patients without receipt of radioiodine», «Reception of RAI therapy» - legend of Table 2 , «the prognosis were better» line 182; sentence in lines 223-224 does not make sense.
2- Manuscript is well organized, with helpful figures and tables.

Experimental design

No comment.

Validity of the findings

1- Authors should explain the fact that 7% of patients in the no-RAI group were not submitted to surgery. Taking into consideration that HCC is only diagnosed in histologic analysis, I wonder how the diagnosis was made in those patients. If no explanation can be found, this issue should be discussed.
2- In the discussion, authors claim that factors associated with increased frequency of RAI therapy may be used in the future to guide the selection of the best candidates for RAI therapy. However, this is a retrospective study aiming at evaluating the survival benefits of RAI in patients with HCC, so the authors cannot claim that those factors should guide our future decisions to perform RAI treatment in patients to HCC. To add to this, performing a total thyroidectomy is a prerequisite to perform RAI treatment in most cases.

Additional comments

In this manuscript, authors evaluated the survival benefit of radioiodine (RAI) treatment in a large dataset of patients with Hurthle Cell Carcinoma of the thyroid gland.
This is a controversial issue, with recent data raising questions on whether HCC patients have poorer prognosis and worse response to RAI when compared with their non-HCC counterparts. Other strengths of the manuscript are: the use of a large well-characterized dataset of patients and the construction of a nomogram for prognosis stratification of patients with HCC, using widely available clinic-pathologic features.
However, authors need to address several points to make the manuscript acceptable for publication. Adding to the points previously mentioned:
1- Authors systematically consider the married status of the patients in the analysis. The relevance of this should be explained in the discussion section.
2- Minor issue: The phrase «vital status» is repeated in the same sentence in lines 86-87.

Reviewer 3 ·

Basic reporting

The text should be improved because in some parts it is not clear and English is not perfect. Please use the list of recommended remarks.

Experimental design

Research question well defined, relevant & meaningful.Rigorous investigation performed to a high technical & ethical standard. Methods described with sufficient detail & information to replicate

Validity of the findings

Improtant findings.
Conclusions should be rewieved.

Additional comments

Quion Yang et al. aim was to find out in large population based research evaluating whether patinents with Hurthle cell carcinoma (HCC) can benefit from postoperative RAI therapy. Their study is interesting. However, there are some issues that have to be clarified before publication in the Peer J.
Major comments:
1. Primary reference should be cited for the sentence of the Introduction (Paolo et al. is not appropriately cited) - lines 44-46
2. Lines 100-115: text is not clear. Can you be more married? Please rewrite.
3. Lines 145-148: State only those that are statistically significant. These are the only one that are important! Omit non-significant factors.
4. Line 191 Refernce Allia at al. is not appropriate (it describes cell cultures and not HCC in human body)
5. Line 202 Add reference Petric R et al. (BMC Cancer 2014, 14:777)
6. Line 223-224: not clear: “It seems that RAI therapy acts absence of survival benefit in the treatment of HCCs patients.”
7. Conclusions in the abstract section and at the end of manuscript should be as follows:
“In summary, this study suggests that RAI therapy was significantly associated with improved OS in patients with Hürthle Cell Carcinoma. However, there were no association between treatment with radioiodine and CSS, possibly due to small number of deaths that were related to HCC.”

Minor remarks
Lines 58-59: Sentence: “Radioactive iodine (RAI) therapy is known widely to be effective in the treatment of postoperative thyroid cancer.” Should be rewritten as follows: “Radioactive iodine (RAI) therapy is known widely to be effective in the treatment of thyroid cancer after thyroidectomy.”
Line 60 : insert “number of ” as follows: “….there are increasing number of studies……….”
Line 64: insert “patients with” as follows: “…..evaluating whether patients with HCC can benefit from……”
Line 127: instead “choose” use “be treated”
Line 218: instead “confirm” use “evaluate”
Line 221: Cox’s
Line 226 and 227: instead of “limited samples” use “incidents”
Line 238: instead: “solid elucidate” use “evaluate”

---

## Round 0.2 · Minor Revisions

Thank you for the modifications introduced in the Ms. The reviewers consider that the Ms is now much more clear and only a few modifications remain. Please address these minor alterations.

·

Basic reporting

The manuscript has been revised and outcomes and stated clearly now.

Experimental design

Research question is well defined and addressed by providing relevant evidence. Grammatical errors have been corrected, and now the manuscript flows much better.

Validity of the findings

Valid findings which will add to the current science knowledge.

Additional comments

Thanks for updating the manuscript as it flows much better now.

Reviewer 2 ·

Basic reporting

This is an improved version of the manuscript, with a better use of English. However, there are a few details that need revision. Here are a few examples:
- Abstract, line 37: the verb is missing in the first sentence (is??)
- Introduction, line 43: «.. is a rare…»
- Resultsl, line 174: «.. seems to be no difference»
- Discussion, line 215-217: sentence needs to be rephrased

Experimental design

No further comments

Validity of the findings

No further comments

Additional comments

This is an improved version of the manuscript, with a better use of English. However, there are a few details that need revision. Here are a few examples:
- Abstract, line 37: the verb is missing in the first sentence (is??)
- Introduction, line 43: «.. is a rare…»
- Resultsl, line 174: «.. seems to be no difference»
- Discussion, line 215-217: sentence needs to be rephrased

Reviewer 3 ·

Basic reporting

Clear and unambiguous, professional English used throughout.
Literature references, sufficient field background/context provided.
Professional article structure, figures, tables are O.K.
Paper is self-contained with relevant results to hypotheses.

Experimental design

Original primary research is within Aims and Scope of the journal.
Research question is well defined, relevant & meaningful. It is stated how research fills an identified knowledge gap.
Iinvestigation performed to a high technical & ethical standard.
Methods dare escribed with sufficient detail & information to replicate.

Validity of the findings

It is the first SEER based study on impact of RAI therapy in more than 2000 patients with Hurthle cell carcinoma. Rationale & benefit to literature is clearly stated. All underlying data have been provided; they are robust, statistically sound, & controlled. Conclusions are well stated, linked to original research question & limited to supporting results.

---

## Round 0.3 · accepted · Accept

Thank you for the modifications introduced in the Ms and congratulations for the acceptance